# Study protocol for a cluster randomised controlled factorial design trial to assess the effectiveness and feasibility of reactive focal mass drug administration and vector control to reduce malaria transmission in the low endemic setting of Namibia

Oliver F Medzihradsky,[1,2] Immo Kleinschmidt,[3,4] Davis Mumbengegwi,[5] Kathryn W Roberts,[1] Patrick McCreesh,[6] Mi-Suk Kang Dufour,[7] Petrina Uusiku,[8] Stark Katokele,[8] Adam Bennett,[1] Jennifer Smith,[1] Hugh Sturrock,[1] Lisa M Prach,[1] Henry Ntuku,[1] Munyaradzi Tambo,[4] Bradley Didier,[9] Bryan Greenhouse,[10] Zaahira Gani,[11] Ann Aerts,[11] Roly Gosling,[1] Michelle S Hsiang[1,2,6]

For numbered affiliations see end of article.

**Correspondence to**
Dr Michelle S Hsiang;
michelle.hsiang@
utsouthwestern.edu

## ABSTRACT

**Introduction** To interrupt malaria transmission, strategies must target the parasite reservoir in both humans and mosquitos. Testing of community members linked to an index case, termed reactive case detection (RACD), is commonly implemented in low transmission areas, though its impact may be limited by the sensitivity of current diagnostics. Indoor residual spraying (IRS) before malaria season is a cornerstone of vector control efforts. Despite their implementation in Namibia, a country approaching elimination, these methods have been met with recent plateaus in transmission reduction. This study evaluates the effectiveness and feasibility of two new targeted strategies, reactive focal mass drug administration (rfMDA) and reactive focal vector control (RAVC) in Namibia.

**Methods and analysis** This is an open-label cluster randomised controlled trial with 2×2 factorial design. The interventions include: rfMDA (presumptive treatment with artemether-lumefantrine (AL)) versus RACD (rapid diagnostic testing and treatment using AL) and RAVC (IRS with Acellic 300CS) versus no RAVC. Factorial design also enables comparison of the combined rfMDA+RAVC intervention to RACD. Participants living in 56 enumeration areas will be randomised to one of four arms: rfMDA, rfMDA+RAVC, RACD or RACD+RAVC. These interventions, triggered by index cases detected at health facilities, will be targeted to individuals residing within 500 m of an index. The primary outcome is cumulative incidence of locally acquired malaria detected at health facilities over 1 year. Secondary outcomes include seroprevalence, infection prevalence, intervention coverage, safety, acceptability, adherence, cost and cost-effectiveness.

**Ethics and dissemination** Findings will be reported on clinicaltrials.gov, in peer-reviewed publications and

## Strengths and limitations of this study

► The study is among the first to evaluate the effectiveness and feasibility of reactive focal mass drug administration in a low-transmission/elimination setting.
► The study is the first to evaluate reactive focal indoor residual spraying insecticide in a low-transmission/elimination setting.
► The 2×2 factorial design allows for evaluation of each intervention individually as well as in combination.
► The study is nested within the current surveillance and response programme of the local Zambezi Region Ministry of Health and Social Services, thus facilitating potential future integration of strategies and creating opportunities for building capacity and infrastructure.
► The Zambezi Region has been subject to recent malaria outbreaks. Such unanticipated shifts in case burden may compromise the capacity of the team to implement the study interventions which are reactive to incident malaria cases.

through stakeholder meetings with MoHSS and community leaders in Namibia.

**Trial registration number** NCT02610400; Pre-results.

## INTRODUCTION

There are presently 35 malaria-eliminating countries worldwide, making active progress towards interruption of domestic transmission and targeting elimination by 2020–2035.[1]



As these countries attain low endemicity levels, malaria transmission clusters spatially and temporally.[2–4] With declining transmission, a higher proportion of infected individuals remain asymptomatic[5–7] and therefore are less likely to present for treatment. This infectious reservoir can perpetuate transmission[8 9] and threaten a country or subnational region's progress towards malaria elimination. Treating these infections may have a greater effect on transmission reduction than focusing exclusively on passively detected symptomatic cases.[10] Reactive case detection (RACD) is a method of actively finding cases among the largely asymptomatic household members and neighbours of passively identified symptomatic index cases. While RACD is widely implemented, there are challenges relating to logistics and acceptability of blood testing. Also, diagnostics in current use such as microscopy and antigen-based rapid diagnostic tests (RDTs) have limited sensitivity for asymptomatic infections, which are generally of low density.[5 9 11–13] There are no controlled studies proving the effectiveness of RACD to reduce transmission. As such, the WHO has recommended against the use of RACD with currently available diagnostics for transmission interruption.[14]

Alternatively, empiric administration of antimalarials to an affected population has been recently endorsed by WHO as a method to interrupt *Plasmodium falciparum* transmission in low-endemicity settings, where this is likely to lead to elimination.[14] By circumventing the operational and technical challenges of testing, mass drug administration (MDA) addresses many of the limitations of RACD. A systematic review examining the impact of over 180 published and unpublished MDA programmes showed that MDA can successfully eliminate *P. falciparum* transmission, though the impact may not be sustained. Interventions showing impact beyond 1 year tended to be in low transmission settings and over 80% (10/12) incorporated additional vector control.[15] There were noted challenges reaching high (over 80%) coverage levels due to logistics, adherence and acceptability. Acceptability is a particular challenge in low transmission settings where community members may not perceive malaria as a threat. The safety and pharmacovigilance of drug administration to large populations is an additional concern as many of those treated are uninfected. Targeted drug administration to smaller populations of individuals at highest risk of malaria, referred to as focal MDA, may mitigate some of the barriers and potential harms of untargeted MDA. A reactive focal MDA (rfMDA) approach, in which household members and neighbours of passively detected index cases are targeted, as is done with RACD, has been used successfully in low transmission *P. vivax* settings in China.[16] RfMDA remains unstudied in *P. falciparum* endemic, low transmission areas of sub-Saharan Africa. RfMDA has the additional advantage of providing prophylaxis to those members of the community who are at highest risk of infection, even if they were not infected at the time of treatment.

While providing antimalarial treatment may address the human reservoir of parasites, vector control measures are necessary to address the mosquito reservoir. Long-lasting insecticide-treated bed net (LLITN) distribution, indoor residual spraying (IRS) of insecticide on the interior walls of homes, larviciding and community education to promote vector avoidance are some commonly used approaches. Due to its proven effectiveness[17] and the seasonal nature of malaria transmission in southern Africa, pretransmission IRS is a widely used vector control method in the region.[18] However, IRS campaigns face spray quality and coverage barriers,[19] particularly with an untargeted 'blanket' approach whereby the goal is usually to reach at least 80% of sleeping structures in the entire endemic area.[20] Resistance may also develop if sprayed insecticides are not periodically rotated.[21] As with strategies targeting the human reservoir, a reactive focal IRS using a highly effective insecticide (reactive vector control or RAVC) may be an effective, operationally feasible, acceptable and cost-saving approach to reducing malaria transmission.

Using a cluster randomised controlled design, this study seeks to evaluate the effectiveness and feasibility of rfMDA and of RAVC, singly and in combination, in comparison to standard of care control interventions in the Zambezi Region of Namibia. Namibia is a low transmission country in southern Africa that, by judicious implementation of IRS, LLITNs and effective case management using RDTs and artemisinin combination therapies (ACTs), reduced incidence from 278 to 9/1000 persons per year between 2001[22] and 2011.[23] RACD, in conjunction with a single yearly round of preseason IRS with dichlorodiphenyltrichloroethane or deltamethrin, has remained the backbone of the country's strategic plan to eliminate malaria by 2020.[24] More recently, the rate of Namibia's incidence decline has plateaued, with outbreaks plaguing the northern border. Given the known limitations of RACD and untargeted seasonal IRS, new strategies may be needed to reach malaria elimination goals.

The antimalarial to be used for rfMDA, as with RACD, is the ACT artemether-lumefantrine (AL). AL is safe and effective in adult and paediatric populations.[25] In Namibia, AL is first-line treatment for uncomplicated cases caused by *P. falciparum*, the species comprising 97% of total malaria infections in the country.[24] In addition to the routine preseason annual IRS administered by the national programme, this study will deploy reactive vector control (RAVC) using the microencapsulated formulation of the organophosphate pirimiphos-methyl (Actellic 300CS; Syngenta AG, Basel, Switzerland). This highly effective insecticide has been shown to have residual bioactivity on most wall surfaces of up to 12 months against susceptible mosquitoes,[26] a low acute toxicity index and is non-teratogenic.[27]

## Aims and objectives

The overall objective of the study is to evaluate the effectiveness and feasibility of two focal, reactive community based malaria interventions targeting the parasite reservoir in humans and the mosquito, each on its own and in

**Table 1** 2×2 factorial study design showing four arms

| | | Reactive and targeted strategies addressing human reservoir | |
|---|---|---|---|
| | | RACD*<br>*2n clusters* | rfMDA†<br>*2n clusters* |
| Reactive and targeted strategies addressing mosquito reservoir | No RAVC<br>*2n clusters* | RACD only arm<br>*n clusters* | rfMDA only arm<br>*n clusters* |
| | RAVC‡<br>*2n clusters* | RACD+RAVC arm<br>*n clusters* | rfMDA+RAVC arm<br>*n clusters* |

*RACD: administering rapid diagnostic test to individuals living in a 500 m radius around an index case; treating positives with artemether-lumefantrine.

†rfMDA: presumptively treating individuals living in a 500 m radius around an index case using artemether-lumefantrine, without testing.

‡RAVC: spraying long-acting insecticide Actellic 300CS to interior walls of living structures of individuals sleeping in a seven household radius around an index case.

RACD, reactive case detection; RAVC, reactive vector control; rfMDA, reactive focal mass drug administration.

combination, in a partnership with the Namibian Ministry of Health and Social Services (MoHSS).

The primary aim is to assess impact of the interventions on cumulative incidence. It is hypothesised that rfMDA and RAVC will each be associated with a 50% reduction in cumulative incidence compared with RACD, and that the combination of rfMDA and RAVC will be associated with a 75% reduction compared with RACD only, that is, assuming an additive effect with interaction. Secondary aims assess effectiveness by comparing seroprevalence and infection prevalence between study arms in an endline cross-sectional survey. Secondary outcome measures of feasibility include coverage, safety, acceptability, adherence, cost and cost-effectiveness. The investigators hypothesise that rfMDA and RAVC will each lead to greater reductions of seroprevalence and infection prevalence than RACD and show at least equivalent feasibility as RACD.

## METHODS AND ANALYSIS

The SPIRIT (Standard Protocol Items: Recommendations for Interventional Trials)[28] recommendations were referenced in developing this protocol.

## Study design

The study is an open label, cluster randomised controlled trial with 2×2 factorial design (table 1) to evaluate the superiority of community-based interventions in response to a passively identified malaria index case. Factorial design allows for comparison of the two study interventions (rfMDA with RAVC) individually, and combined, against reference (RACD only). The factorial design permits each single intervention to be compared with the reference in *2n* clusters per arm, while the combination of the two interventions is compared with the reference in *n* clusters per arm. The study design has been generated through an iterative process of engagement with the national and regional MoHSS directorship and with Zambezi community leaders.

## Study setting and trial preparations

More than two-thirds of Namibia's 2.51 million population[29] live in its northern malarious regions, characterised by low to moderate endemicity. The tropical conditions and proximity to Angola and Zambia elevate the risk of persistent transmission due to imported malaria.[30 31] Malaria is almost entirely due to *P. falciparum*.[24] Following the wet season, the high transmission season typically begins in December, peaks between January and April and ends in May. The study area is in Zambezi Region, located in the northeastern part of the country (figure 1, inset) and consists of the catchment areas of 11 health facilities in western Zambezi Region (figure 1), covering approximately 8000 square km with an estimated population of 35 381, based on 2011 census data[32] and estimated population growth rate. The Zambezi Region was selected as a study site due to having a sufficiently high incidence to provide power to answer the study questions, while still representing a very low transmission epidemiology, defined by WHO as areas with an annual parasite incidence of fewer than 100 cases per 1000 population.[33]

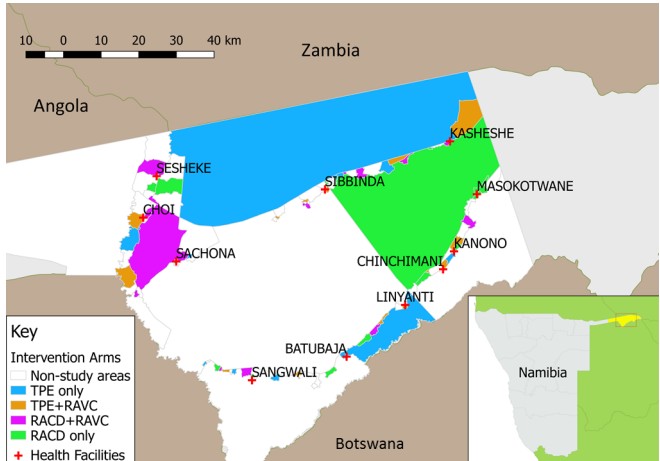

**Figure 1** Study area in western Zambezi Region, Namibia, showing intervention areas and non-study areas. RACD, reactive case detection; RAVC, reactive vector control.

Prior to the trial, various activities were undertaken to facilitate study implementation and gather baseline epidemiological data. A geographic reconnaissance (GR) survey was conducted to geolocate and assign all sleeping structures in the study area with a unique GR code. A sticker with the GR code was placed in the doorway of each sleeping structure and in the health passports of all residents. A tablet-based system for rapid reporting of malaria cases and their GR code was established in study health facilities. A centralised spatial decision support system (SDSS)[34] was also established to receive data from the rapid reporting system and facilitate spatial planning for the study interventions. Paper-based malaria registers from western Zambezi health facilities for 2013 and 2014 were reviewed to calculate annual baseline incidences to the level of the village. Incidence data from 2015 were not available at the time of study planning because of transition to the newly established rapid reporting system, which was used to capture 2016 incidence data. A cross-sectional survey was conducted in the catchment areas of six randomly selected health facilities during April to June 2015 to measure prevalence of infection as well as coverage of interventions such as bed nets, preseason IRS and access to healthcare. In 2016, coverage of the annual preseason IRS was monitored using mSpray technology.[35] Community sensitisation took place prior to the trial launch and consisted of meeting with MoHSS regional leadership and engaging with local community leaders, health workers and villagers in meetings and via radio announcements.

## Randomisation

The unit of randomisation is a census enumeration area (EA). Of 102 EAs in the study area, 56 were selected for inclusion in the trial, excluding EAs without incident cases in the prior 3 years. These clusters were randomly allocated by computer-generated algorithm to one of four arms (figure 1) using restricted randomisation to balance the distribution of key characteristics across study arms, including EA level incidence in 2013 and 2014, population size, population density and healthcare access as measured by mean household distance to a health facility. A set of 100 000 reassignments meeting the restriction criteria was generated. A set of 10 allocations meeting the restriction criteria was randomly selected by computer algorithm. The final allocation was randomly selected by local MoHSS staff.

## Procedures

### Index case enrolment at health facility

Index case enrolment and subsequent procedures are shown (figure 2). Each study intervention will be triggered by an index case diagnosed by RDT (CareStart Malaria HRP2/pLDH(Pf/PAN) Combo, Access Bio, Somerset, New Jersey, USA) or microscopy[36] at one of the 11 study health facilities and reported through the rapid reporting tablets. Index cases detected at the regional referral hospital or five private health clinics may also

trigger a study intervention, with these cases reported via a telephone call to study staff or detected through weekly visits by study staff. Health facility staff will administer the MoHSS case investigation form to RDT positive index cases and obtain consent for a second pretreatment fingerprick to collect a dried blood spot (DBS) for subsequent confirmatory molecular testing. This questionnaire will include basic information such as demographics, occupation, travel, use of vector control interventions, recent treatment, GR code and village of residence if GR code not available. The health facility worker will notify the index case that a team may visit their home to conduct an additional questionnaire and offer malaria interventions to household members and neighbours. RDT-positive cases will be provided with antimalarial treatment at the health facility per national guidelines.[36]

### Enrolment of participants for study intervention

For index cases meeting criteria to trigger a response (table 2), their household and neighbours will be eligible for the study intervention. Using output from the SDSS to plan the intervention, a study team, consisting of a field investigator, nurse and driver/data collector, will be dispatched to the home of the index case. In the event that there are more pending intervention events than available study staff, index cases in villages with the highest incidence will be prioritised. After establishing contact with the village headman, the study team will investigate the index case, consisting of: verification of information collected in the rapid reporting system; additional questions regarding occupation, travel history and use of vector control interventions; capture of GPS coordinates and collection of DBS if not drawn at the health facility. Study teams will then enrol eligible individuals (table 2) in the target area and deploy the intervention to which the associated EA had been assigned. For rfMDA and RACD, the target area consists of the index case household and surrounding households within 500 m, with a minimum enrolment of 25 individuals. If fewer than 25 consenting participants live within 500 m of the index case, enrolment will stop at this distance for that target area. For RAVC, the target area will encompass the seven households in closest proximity to the home of the index case, including the index case household. Given malaria risk is associated with proximity to an index case household,[3 37 38] study teams will approach households with progressively increasing distance from the index household. In order to maximise enrolment and coverage, teams will visit each target area at least twice, at different times of the day and preferably within two and no longer than 5 weeks of the index case presentation date. On the second visit, the team will first revisit households with previously missed residents, prior to moving on to the next closest household from the index case household. Prior to approaching individuals within each household, a household level questionnaire will be

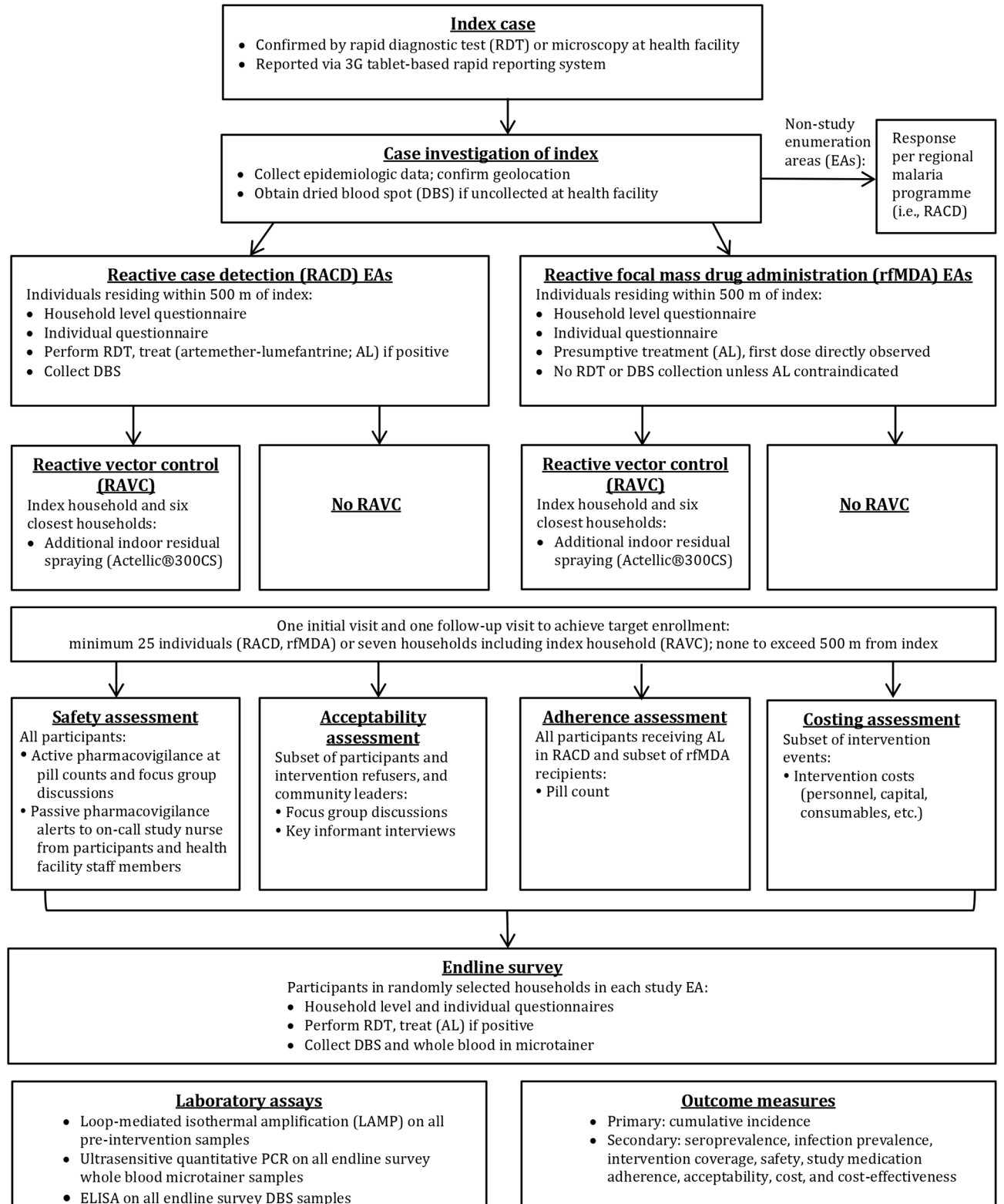

**Figure 2** Flow of study procedures.

conducted with the household head or their representative to determine the household size, demographic characteristics of household members, construction materials of the home, coverage of LLITNs and preseason IRS and to develop a next-visit plan to enrol individuals not present that day.

**Reactive case detection**

A questionnaire similar to that used in routine index case investigation will be administered to all eligible consenting individuals in a private area where possible in order to maintain confidentiality. Blood will be collected by fingerprick for CareStart Pf/PAN RDT and

**Table 2** Inclusion and exclusion criteria for interventions

|  | Inclusion criteria | Exclusion criteria |
|---|---|---|
| Index case triggers a reactive intervention | ▸ Index case with RDT or microscopy confirmed malaria identified passively at a study health facility<br>▸ Triggering index case resides (or stayed≥1 night within prior 4 weeks) in study enumeration area | ▸ Triggering index case diagnosed by active case detection |
| Receipt of test (rfMDA or RAVC) or control (RACD) intervention | ▸ Individual resides (or stayed≥1 night within prior 4 weeks) in study enumeration area within 500 m of home of triggering index case (RACD or rfMDA)<br>▸ Household located in study enumeration area among six closest households to triggering index case household (RAVC) | ▸ Study intervention already implemented in individual's household during the prior 5 weeks (RACD or rfMDA)<br>▸ Study intervention already implemented in individual's household during current transmission season (RAVC)<br>▸ Household sprayed by MoHSS with DDT or deltamethrin in the past 24 hours<br>▸ Household head refusal to participate (RAVC)<br>▸ Individual level refusal to participate (RACD or rfMDA)<br>▸ Refusal of RAVC is not an exclusion criterion for RACD or rfMDA.<br>▸ Refusal of rfMDA or RACD is not an exclusion criterion for RAVC |
| AL administration | ▸ Meets above inclusion criteria for rfMDA or RACD | ▸ Reported pregnancy in first trimester<br>▸ Prior regular menstruation followed by amenorrhea for most recent 4 weeks and refusal of pregnancy testing or positive pregnancy testing<br>▸ Weight under 5 kg<br>▸ Age under 6 months<br>▸ Severe/complicated malaria (based on clinical assessment)<br>▸ Prior allergy to AL<br>▸ Personal history of cardiac dysrhythmia<br>▸ Family history of long QT syndrome<br>▸ Regular intake of specified QT-prolonging medications* |

*Cardiac antiarrhythmic, neuroleptic, tricyclic antidepressant, prokinetic and antiemetic gastrointestinal, second generation antihistamine, opioid (methadone) and antimicrobial (macrolide and fluoroquinolone antibiotics, triazole antifungals) agents.
AL, artemether-lumefantrine; DDT, dichlorodiphenyltrichloroethane; MoHSS, Ministry of Health and Social Services; RACD, reactive case detection;  RAVC, reactive vector control; RDT, rapid diagnostic test; rfMDA, reactive focal mass drug administration.

DBS. Any participant found to be RDT positive will be assessed for antimalarial eligibility (table 2) and treated with AL for uncomplicated cases. Beginning October 2016, the MoHSS introduced single low dose primaquine for its antigametocidal effect,[39] which study nurses will administer to participants with uncomplicated malaria. Primaquine dosing will be weight banded to approximate the target of 0.25 mg/kg and administered in conjunction with the first AL dose. Exclusion criteria consistent with national guidelines[36] include pregnancy, age under 1 year, breastfeeding for 12 months or fewer and allergy to primaquine.

### Reactive focal mass drug administration
After administration of the same individual-level questionnaire as in RACD, participants will be screened for

presumptive antimalarial treatment with AL. Eligible participants (table 2) will be weighed and, without blood testing, receive two times per day AL (unit doses 20/120 mg, 40/240 mg, 60/360 mg and 80/480 mg for weights 5–14 kg, 15–24 kg, 25–34 kg and ≥35 kg respectively), administered 8 hours apart on day 1 and 12 hours apart on days 2 and 3.

Prior to taking the first AL dose by directly observed therapy, each participant will receive instructions on self-administration of the remaining five doses and will be required to restate correctly the administration instructions for self and any enrolling minors. Participants will be requested to retain AL packaging for a potential a return visit pill count. Individuals in rfMDA clusters with contraindications to AL (table 2)

will be offered RDT testing. RDT-positive individuals in all arms with AL contraindications and those with severe malaria or other illnesses requiring medical attention will be referred to the nearest health facility for care. Participants receiving rfMDA will not be given primaquine, as effects on safety remain an area of ongoing research.[40]

The first trimester (weeks 0–14) of pregnancy is a contraindication to AL per MoHSS national policy.[36] Female participants acknowledging first trimester pregnancy will be offered an RDT that, if positive, will result in referral to the nearest health facility for alternative treatment with MoHSS-endorsed quinine. Study nurses will enquire about menstrual history on all females over 10 years of age. Postmenarchal females reporting new amenorrhea for the preceding 4 weeks will be offered pregnancy testing. Refusal of pregnancy testing will constitute an AL exclusion and indication for health facility referral.

### Reactive vector control
In the RAVC clusters of the study, Actellic 300CS will be sprayed on ceilings and walls of all sleeping structures, adhering to the national spray application protocol, after obtaining consent and requesting residents to reposition household content. No room containing human or animal inhabitants will be sprayed. Teams will instruct inhabitants to rinse floors of sprayed structures with water before allowing children re-entry and to refrain from tampering with sprayed walls until after the malaria season. Vector susceptibility to the insecticides used in MoHSS preseason IRS and in RAVC as well as residual bioefficacy of the insecticide on sprayed walls, will be measured as described below.

### Pill count
Participants prescribed AL in RACD arms and participants from one randomly selected target area in each rfMDA EA will receive a postintervention follow-up visit to assess adherence. At this encounter, the retained AL blister packs will be inspected and remaining doses recorded. The targeted timing of this pill count visit will be 7–10 days after AL was initiated, with a maximum 30-day interval.

### Acceptability assessment
Acceptability will be assessed according to refusal rates and questions in the endline survey addressing participants' hypothetical willingness to accept the same intervention in the future. A qualitative assessment will also be conducted through focus group discussions (FGDs) in each study arm, consisting of conseting study participants and refusers, in order to assess community acceptability of the interventions. Trained facilitators will explore positive/negative intervention experiences, reasons for opting in/out and perceived malaria risks. FGDs will be gender segregated, include youths (15–17 years old) and adults separately and exclude community leaders. Individual interviews will be held with key informants (KIs)

consisting of male and female leaders at the village, constituency and regional governmental levels. Trained interviewers will inquire about the KI's role in the trial, acceptability of the interventions and experience with elimination efforts in Namibia. FGDs and KI interviews will be audio-recorded with consent.

### Endline survey
A cross-sectional endline survey will take place at the end of the 2017 transmission season to measure seroprevalence and infection prevalence. Study teams will visit randomly selected households in each study EA. Household and individual level questionnaires similar to those used during the interventions will be administered to collect demographic and epidemiological data. A fingerprick will be used to collect blood for a CareStart Pf/PAN RDT test, DBS and 250 uL whole blood in a microtainer for subsequent serologic and molecular assays. Treatment for CareStart RDT-positive individuals will be provided per national guidelines.

### Costing and cost-effectiveness assessment
Study teams will compile expenditure data (including consumables, equipment and overhead, labour and estimated values of donated goods) from ledgers, budget and staff interviews.

### Consent
Except for RAVC, for which informed consent will be obtained from the household head, all other procedures will require individual level written informed consent. Consent will be requested separately for each study procedure (eg, questionnaire, blood testing, drug administration) and at each encounter (eg, primary study interventions, endline cross-sectional survey, acceptability assessment). Facilitators will obtain verbal informed consent for focus group discussions and KI interviews and written consent for all other procedures. Informed consent may be collected from all members of a given household at once. Consent for minors less than 18 years of age will be obtained from a parent or guardian and minor assent will also be obtained from participants 12–17 years old. Informed consent will be conducted in SiLozi, the predominant local dialect, or English, the national language.

### Non-trial care
Participants will be free to seek usual and as-needed medical care at their own discretion, with no effect on study eligibility or arm allocation. Participants undergoing blood testing or receiving medication(s) or insecticide spraying will receive anticipatory guidance on potential side effects. In the event of symptoms, participants will be instructed to notify the on call study nurse, who will be available at all hours and to seek appropriate medical care. Study teams encountering individuals with severe or complicated malaria, uncomplicated malaria meeting an exclusion criterion for AL, non-falciparum malaria requiring radical cure or another need

for further evaluation will refer such participants to the nearest health facility.

## Entomological monitoring

Susceptibility of the vector population to all four classes of insecticide will be assessed at baseline and again at the conclusion of the study in each of the four arms. Adult *Anopheles*, reared from larvae and the F-1 progeny of wild females, will be tested using standard WHO susceptibility tests.[41] Mosquitoes will be identified by microscopy and the numbers of identified species recorded. *Anopheles gambiae* complex and the *An. funestus* group will be identified by PCR.[21] WHO cone bioassays[42] will be conducted on a sample of walls within 1 month of spraying, using insectary reared susceptible mosquitoes. If sufficient mosquitoes are found and if there is evidence of resistance, samples will be exposed to different concentrations of insecticide on filter paper and mortality will be assessed 24 hours postexposure to obtain $LC_{50}$ data or exposed to the standard WHO diagnostic dose for varying lengths of time to estimate $LT_{50}$s.

## Data management

Questionnaires will be programmed using the Open-DataKit platform,[43] with data entered directly into password-secured tablets. Questionnaires will be coded with checks for internal consistency. Data quality will be reviewed by data managers with necessary feedback communicated to teams in real-time and in weekly meetings. Tablets will be stored in a locked location when not in use. Data will be uploaded daily to an encrypted cloud-based server. Audio-recordings from FGDs will be translated, transcribed and uploaded into ATLAS.ti (Scientific Software Development GmbH, Berlin, Germany).

To facilitate follow-up visits and linking between datasets, data with the exception of transcribed FGDs will contain personal identifiers. Only the data managers and principal investigators will have access to identified data. Data analysts will use anonymised data. On completion of the trial and publication of findings, data will be stored for at least 5 years in secure databases. Authorised representatives of the sponsor, ethics committees or regulatory bodies may inspect and audit all documents and records on request.

## Laboratory methods

All DBS will be collected onto Whatman 3 MM paper (Sigma-Aldrich, St. Louis, Missouri, USA), allowed to air dry and transported in sealed plastic bags with desiccant. These will be stored at 4°C within 24 hours and at −20°C within 1 week. DNA will be extracted using the saponin/Chelex method.[44] In light of reports demonstrating faster processing time and comparable sensitivity relative to conventional PCR,[45–47] loop-mediated isothermal amplification (LAMP) will be used to test samples collected in RACD or rfMDA. LAMP will first be performed using genus-specific (Pan) primers (Eiken Chemical, Tokyo, Japan) followed by *P. falciparum*-specific primers on

Pan positive samples. For quality assurance, nested PCR targeting the *P. falciparum cytochrome b* gene[48] will be performed on all LAMP positive samples and a subset of negatives.

To evaluate the secondary outcome of seroprevalence, ELISA will be used to measure antimalarial antibody titers among individuals in the endline survey. Antibodies will be extracted from DBS, and in the case of the endline survey from whole blood, using previously described methods.[49 50] Seroassays will target the blood stage antigens merozoite surface protein 1 and apical membrane antigen 1, both validated biomarkers of *P. falciparum* exposure.[50] Other antigens sensitive and specific for recent exposure are undergoing current evaluation[51] and may also be used. ELISA assays will be performed in duplicate and optical densities recorded. Other serologic platforms, including fluorescent bead array and protein microarray, may be used to analyse responses to multiple antigens. To evaluate the secondary outcome of infection prevalence, all endline survey samples will undergo an ultrasensitive method of *P. falciparum*-specific quantitative PCR,[52] using the cell pellet from 100 μL whole blood per sample, to measure parasite density.

All molecular and serological assays will be performed solely for malaria research purposes and will have no impact on the clinical management of study participants.

## Outcomes and measures

The primary outcome will be cumulative incidence of RDT or microscopy confirmed locally acquired cases, identified passively at health facilities and reported through the rapid reporting system. Determination of local acquisition will be based on self-reported lack of travel in the previous 8 weeks. Due to the lifecycle of *P. falciparum* in the mosquito and the distribution of distinct transmission chains across space and time, an EA level impact of the intervention on transmission would not be expected for at least several weeks after the first intervention for an EA. Thus, cumulative incidence will be calculated based on a delayed window of 8 weeks after the first index case reported in each study EA during the study period. As secondary outcome measures of effectiveness, seroprevalence will be calculated using ELISA and infection prevalence by ultrasensitive quantitative PCR. Secondary outcomes of feasibility will include coverage, safety, acceptability, adherence, costs and cost-effectiveness. Further details of outcome measures are outlined in table 3.

## Sample sizes and power calculations

Based on pretrial malaria incidence in 2013 and 2014, the study was originally powered for an expected incidence of 12.5/1000 persons per year in the RACD arm. Due to an unexpected outbreak of malaria cases during the pilot phase in 2016, the study team was unable to adhere to the protocol of responding to all incident cases in study EAs. Staffing was subsequently scaled up and clusters restrictively rerandomised as described above. The trial

**Table 3** Method of outcome measurements

| Outcome | Measurement metrics |
| --- | --- |
| Incidence | Cumulative incidence of RDT or microscopy confirmed locally acquired cases identified at study health facilities |
| Seroprevalence | Prevalence of antibody response to validated *Plasmodium falciparum* antigens detected by ELISA |
| Infection prevalence | Prevalence of infection detected by ultrasensitive qPCR using whole blood |
| Intervention level coverage | Proportion of index cases for which a study intervention is implemented |
| Individual level coverage | RfMDA: Proportion of rfMDA-eligible residents of target area who take the first dose of AL<br>RACD: Proportion of RACD-eligible residents of target area who undergo fingerprick for RDT testing<br>RAVC: Proportion of RAVC-eligible households in target area that are sprayed with Actellic 300CS |
| Safety | Frequency of serious adverse events[51] deemed possibly, probably or definitely related to study intervention |
| Acceptability | Quantitative assessment measured by proportion of eligible individuals who consent to receive the assigned intervention and proportion of participants sampled in endline survey who indicate they would participate in the intervention again if offered<br>Qualitative assessment in focus group discussions and key informant interviews |
| Adherence | Among those selected for pill count, proportion of participants found to have completed AL course |
| Cost | Cost (materials, labour and infrastructure) per intervention event and per person enrolled, and in RACD arms, cost per additional infection found |
| Cost-effectiveness | Cost per case averted and incremental cost effectiveness ratio for rfMDA versus RACD, RAVC versus no RAVC and rfMDA+RAVC versus RACD alone (should the test interventions prove to be more effective than control) |

AL, artemether-lumefantrine; qPCR, quantitative PCR; RACD, reactive case detection; RAVC, reactive vector control; RDT, rapid diagnostic test; rfMDA, reactive focal mass drug administration.

was relaunched in January 2017 for a planned 12 months of primary outcome measure data collection. Revised power calculations are based on 2016 regional incidence (32.5/1000 persons), study EA total population (18,022) and study EA harmonic mean population (276). Hypothesised effects on incidence of RACD, rfMDA, RAVC and rfMDA+RAVC are 25%, 50%, 50% and 75%, respectively. The effective sample size of the population is 15 456, predicting 206 enrolled index cases and 5150 total enrolled individuals. Allowing for refusals (5%) and contraindicated interventions due to an antecedent intervention in the prior 5 weeks (10%), a total enrolment of 4403 individuals is targeted. Using these assumptions and sample sizes and a coefficient of variation of 0.95 calculated from baseline EA level incidence rates, the trial will have at least 80% power to detect the hypothesised effects of rfMDA versus RACD, RAVC versus no RAVC and rfMDA+RAVC versus RACD alone.

The endline survey will sample 25 randomly selected households in each study EA. An estimated 6300 sampled and 5040 enrolled individuals are anticipated, assuming a mean household size of 4 people and 20% non-response. With a sample size of 2520 in the two rfMDA arms and 2520 in the two RACD arms and the same numbers for the two RAVC based and two non-RAVC arms, the survey will have 80% power to detect a minimum detectable decrease in seroprevalence to 5.3% (47% reduction) for rfMDA versus RACD and RAVC versus non-RAVC,

assuming 10% seroprevalence in the RACD arms.[50] As infection prevalence is expected to be low, power calculations were not be performed for this secondary outcome measure.

With regard to the feasibility evaluation, the trial will aim to achieve 80% coverage based on modelling studies[10] and experience from successful presumptive treatment programmes[15] as well as 80% adherence among subjects receiving AL as part of the study intervention. If the estimates for incidence and target area population are correct, the study will be powered to measure at least 80% coverage for the rfMDA, RACD and RAVC interventions as well as 80% adherence (Alpha=0.05). Power calculations were not performed for the safety assessment as few to no serious adverse events[53] (SAEs) are anticipated. For the acceptability assessment, a total of 12 FGDs, three per study arm (adult females, adult males and youths) consisting of 8–12 participants each, will be conducted along with at least six KI interviews. In the costing evaluation, detailed expenditure analysis will be performed for 10 target areas per study or control intervention.

### Statistical analysis

One-way frequency tables for all categorical variables and distributions, ranges and outliers for continuous variables will be generated to perform range checks, quantify the amount of missing data and generate descriptive findings characterising EA and target area characteristics. These

analyses will be stratified by intervention (ie, rfMDA vs RACD and RAVC vs no RAVC). The baseline group covariates will be analysed for equality (eg, by Rao-Scott $\chi^2$). Although we expect randomisation to produce balanced covariate structures, we will consider methods of adjustment to balance baseline covariates (eg, LLITN use, pretransmission season IRS coverage, travel history, housing, occupation, ecological factors) should there be differences between the arms. If an imbalance is found, a second set of models will be run for all primary analyses. Adjusted models will be used to reweight the data, either through inverse probability weighting or targeted minimum loss based estimation, in order to estimate effects based on equally distributed covariates.

Three comparisons of cumulative incidence will be made using an intention-to-treat (ITT) approach: rfMDA versus RACD arms (28 EAs each), RAVC versus non-RAVC arms (28 EAs each) and rfMDA+RAVC versus RACD alone (14 EAs each). Outcomes will be assessed at both the level of the EA (cluster) and the individual, adjusting for clustering. The modelling approach for individual level analyses will adjust for observation correlation by EA for analyses using the overall sample and by both EA and target area for analyses using the target area sample. Generalised estimating equations (GEEs) will be used to perform the proposed primary analyses. GEE methods account for the correlation of persons within clusters and EAs. Robust Huber-White 'sandwich' SEs will be used to obtain correct inferences even if the chosen correlation structure remains slightly mis-specified. Alpha will be set at 0.05 for all planned comparisons. Primary analyses will follow an ITT model. Individuals will be analysed according to their randomisation arm regardless of received treatment. Additional analyses will consider adjustments for variability in intervention timing and coverage, differential administration of single low dose primaquine in RACD versus rfMDA arms, proximity to the nearest household receiving an alternate intervention and variation in EA characteristics that differed by intervention arm, including receipt of RAVC and/or rfMDA during the pilot phase of the study.

Proportional hazards regression for an individual's survival to the first episode of malaria will also be performed. Individual incident cases registered in the rapid reporting system will be tallied by EA. Using summary population data from the GR for each EA, the number of individuals in each EA who did not have an incident case of malaria reported will be calculated and a record generated for each hypothesised non-infected individual. These will be combined with the records for incident cases to create a dataset with one record for each individual in every EA. This dataset will be used to run a Cox hazard model of time to either incident infection or the date of dataset aggregation. The start time for each EA will be the date of the first incident infection, that is, the index infection that triggered a study intervention. Hazard models will adjust for clustering of observations by EA. Significance will be calculated using a robust 'sandwich' estimator. To consider different criteria for intervention effectiveness, three models will be run: all cases postdating the index case; all cases excluding those with travel history (ie, presumed imported cases) and all cases censoring those with either a travel history or a detection date within 8 weeks of the index case.

Although power is limited by the constraints of the total number EAs in the study area, summary measures of incidence and intervention effects will be described for certain subgroups of interest within the study population (eg, members of each gender, children and those living in highest-incidence geographic areas).

Secondary outcome measures will be assessed descriptively or as appropriate, using GEE techniques to account for the clustered nature of the data collection. For the qualitative analysis, transcripts of FGDs and KI interviews will be coded in ATLAS.ti5. Major themes will be identified by two members of the study team independently. Any discrepancies will be resolved by inviting a third member of the study team to verify findings. Data will then be analysed to provide a description of the barriers and facilitators to acceptance. For the costing assessment, calculations will include total costs of the study interventions, costs per intervention event and cost per individual enrolled. For the cost-effectiveness assessment, the main outcome measures will be cost per case averted and incremental cost effectiveness ration, should the test interventions prove to be more effective than the control interventions.

## ETHICS AND DISSEMINATION
### Monitoring, harms and auditing
After initial training, each study team will continue to undergo regular evaluation and supervision of its enrolment and intervention activities, by the Site Manager, Research Manager and designated team leaders, to ensure compliance with study protocol. The Data Manager and Research Manager will carry out weekly internal audits of completed questionnaires and case investigation records to ensure high-quality data capture and provide team member feedback.

Prior to launch, trainings will be held at each study health facility to educate staff on the study interventions and on the spectrum of potential toxicities due to AL, Actellic 300CS and in the event of national rollout, single low dose primaquine. In addition to passive pharmacovigilance, study nurses will proactively inquire about symptoms among participants during return visits to a target area for further enrolment or pill counts. Study nurses will review all SAEs within 48 hours and perform a follow-up visit within 72 hours in conjunction with the study physician. The physician will assign causality, recommend retention versus exclusion from the study and report the SAE to the investigators. All SAEs will be reported to the Data and Safety Monitoring Board (DSMB) and to the MoHSS Therapeutics Information and Pharmacovigilance Center in Windhoek, Namibia, within 7 days

of occurrence. Reports of all SAEs classified as possibly, probably or definitely related to a study intervention, as well as of all suspected unexpected serious adverse reactions (SUSARs), will be submitted to the institutional review boards of the participating institutions, within 7–10 days of occurrence or per the boards' stipulations.

The DSMB will consist of at least four content experts in malaria elimination, clinical trial design and pharmacology who did not contribute to the study's design and are independent of the project's operations and external to the investigators' institutions. The DSMB will review the protocol and analytical plan prior to launch. During the study, the DSMB will examine SAE and SUSAR reports and have final authority on such participants' continuation versus withdrawal from the study. The DSMB will meet every 3–4 months to review trial safety and enrolment data and vote to recommend trial continuation versus early termination. Stopping guidelines will be based on the prevalence of SAEs deemed to have a causal association with the study interventions as well as enrolment and consequent likelihood of endpoint assessability.

The study's Steering Committee will consist of senior representatives of the MoHSS, University of Namibia, University of California, London School of Hygiene and Tropical Medicine, Clinton Health Access Initiative and Novartis Foundation and will provide technical oversight of the study. The Committee will assist with interpretation, dissemination and extrapolation of findings for international impact while ensuring consistency with MoHSS goals for elimination. The Committee will meet once yearly in Namibia, participate in teleconferences one to two additional times a year and have ongoing interactions with study team members on an as needed basis.

## Dissemination

Study findings will be shared in stakeholder meetings attended by regional and national MoHSS representatives, with health facility staff and community leaders, through peer-reviewed publications and at scientific conferences. Results will also be reported on the Web through clinicaltrials.gov.

**Author affiliations**
[1]Malaria Elimination Initiative, Global Health Group, University of California San Francisco, San Francisco, California, USA
[2]Department of Pediatrics, Benioff Children's Hospital, University of California San Francisco, San Francisco, California, USA
[3]Department of Infectious Disease Epidemiology, The London School of Hygiene and Tropical Medicine, London, UK
[4]Faculty of Health Sciences, School of Pathology, University of Witwatersrand, Johannesburg, South Africa
[5]Multidisciplinary Research Centre, University of Namibia, Windhoek, Namibia
[6]Department of Pediatrics, University of Texas Southwestern Medical Center, Dallas, Texas, USA
[7]Division of Prevention Science, University of California San Francisco, San Francisco, California, USA
[8]National Vector-borne Diseases Control Programme, Ministry of Health and Social Services, Windhoek, Namibia
[9]Clinton Health Access Initiative, Boston, Massachusetts, USA
[10]Division of Experimental Medicine, Department of Medicine, University of California San Francisco, San Francisco, California, USA
[11]Novartis Foundation, Basel, Switzerland

**Acknowledgements** The authors would like to thank the following individuals for their contributions to the study protocol development: Edith Roset Bahmanyar, Christopher Lourenco, Mwalenga Nghipumbwa, Agnes Mwilima, Griffith Siloka, Gerard Kelly, Phil Rosenthal, Kim Baltzell, Nelago Indongo, Deepika Kandula, Deepa Pindolia, Arnaud Le Menach, Andy Tatem, Nick Ruktanonchai, Ronnie Bock, Tatiana Hans Angula, Richard Feachem, Kenneth Matengu and Richard Kamwi.

**Contributors** MSH, IK and RG conceptualised and developed the study and its design. DM, AB, JS, HS, M-SKD, KWR and OFM contributed to study design. All authors contributed to protocol development. KWR, MSH, M-SKD and OM wrote the protocol. OM, MSH and IK wrote the manuscript. All authors approved the manuscript final draft.

**Funding** This work is financially supported by Novartis Foundation [A122666], the Bill & Melinda Gates Foundation [OPP1138299] and the Horchow Family Fund [5300375400]. Cofunding for drugs, RDTs and other existing infrastructure was also provided by the MoHSS.

**Competing interests** AA and ZG are employed by Novartis Foundation. All other authors declare no competing interests.

**Patient consent** Not required.

**Ethics approval** The protocol has been approved by the Ministry of Health and Social Services of Namibia, University of Namibia Centre for Research and Publications, University of California San Francisco Human Research Protection Program (HRPP; #15-17422) and The London School of Hygiene and Tropical Medicine Observational/Interventions Research Ethics Committee (#10411). This trial is registered at clinicaltrials.gov (NCT02610400, Unique Protocol ID: OPP1089413_Namibia_TPE).

**Provenance and peer review** Not commissioned; externally peer reviewed.

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
