## [Reviewer comments · BMJ Open]

ARTICLE DETAILS

TITLE (PROVISIONAL)	A study protocol for a cluster randomised controlled factorial design trial to assess the effectiveness and feasibility of reactive focal mass drug administration and vector control in the low transmission setting of Namibia
AUTHORS	Medzhradsky, Oliver; Kleinschmidt, Immo; Mumbengegwi, Davis; Roberts, Kathryn; McCreesh, Patrick; Kang Dufour, Mi-Suk; Uusiku, Petrina; Katokele, Stark; Bennett, Adam; Smith, Jennifer; Sturrock, Hugh; Prach, Lisa; Ntuku, Henry; Tambo, Munyaradzi; Didier, Brad; Greenhouse, Bryan; Gani, Zaahira; Aerts, Ann; Gosling, Roly; Hsiang, Michelle

VERSION 1 – REVIEW

REVIEWER	Joseph Pryce Liverpool School of Tropical Medicine, United Kingdom
REVIEW RETURNED	27-Sep-2017

GENERAL COMMENTS	This is a well-designed protocol that clearly defines the issues with the current standard intervention, the purpose of the study and the methodology that will be used. It will be a very large and complicated study. In addition to the testing the efficacy of two novel interventions, the study includes a qualitative investigation into the acceptability of the interventions, a cost-analysis, vector data collection and even a diagnostic test accuracy component. The authors have done a good job detailing the procedures that will be undertaken in each of these work-streams, and provide clear rationale for the study design and interventions that have been chosen. I have a few comments and would suggest only a few minor revisions; these are based on my experience producing systematic reviews of randomised controlled trials for both malaria treatment and vector control interventions. Background: 1.5 Drug Administration • My understanding is that single dose PQ is provided to all Pf index cases in both arms. It is also provided to RDT-positive household members and neighbours in the RACD arm, but not presumptively to household members and neighbours in the TPE arm. It seems there is a possibility for a confounding effect of PQ (if it is hypothesised that PQ has potential to block transmission) – skewing the results in favour of RACD. I assume there is a safety-related problem with
---

providing PQ in the TPE arm to healthy people?
Perhaps this could be explained. The issue should also be acknowledged/adjusted for if possible.

- On that note is there an issue with PQ at this dose causing haemolytic anaemia in people with G6PD deficiency? Is this a risk that needs to be considered in the RACD arm? Perhaps for inclusion under the safety and risks section.
- “Anyone who is excluded from TPE receives an RDT instead”. It would be useful to state how these people will be analysed, presumably they are still considered under TPE on an intention to treat basis.

Study Site:

- It states in the background that previous trials of TPE were challenged by their high transmission settings, but it appears here the Zambezi region has been selected for being the most malarious region in the country. I thought the purpose of the study was to trial TPE in a low transmission setting so the reasons behind the choice may need slightly further clarification.

Study design:

5.1 Overview

- Table 2. Reactive Vector Control exclusion criteria states households sprayed by MoHSS in past 24 hours will be excluded. Does this mean houses that have been sprayed only 2 days previously will be sprayed again? It also states later in the paper that households that have received RAVC in that season will not be sprayed again; this could be added to this table.

Study procedures:

6.1 Randomisation

- It states that after removing the Enumeration Areas that do not meet the inclusion criteria, there was a total of 54 EAs. I am unclear how this list increased to 56 so that the randomisation could produce equal numbers in each arm. Where did the extra two EAs come from?
- I assume the allocation is randomly selected using a computer randomised algorithm – it is useful to state this as describing the randomisation process is an important part of reviewing the risk of bias in a systematic review.
- I am not sure if the interventions were applied during year 1 of the study. If so, there is a risk of contamination from the study villages that received RAVC as pirimiphos is active for up to 12 months. EAs in the non-RAVC arm may therefore be benefitting from IRS from the previous season. It may be possible to consider a measure of time since last IRS treatment as a covariate during the analysis, or discuss the limitations of the study for reviewing the impact of RAVC vs a control group that is potentially contaminated by IRS (both as a standard MoHSS procedure and in those that received RAVC in the first year of the study)

Entomological management:

9.2 Mosquito identification

- I note that the study does not intend to report entomological outcomes, but if mosquitoes are being identified, it would be interesting to know how these mosquitoes will be collected. If this is conducted in a standardised way eg. with CDC light traps, it would be possible to report the number of vectors per trap per night (or time period).

	This measure of vector density is a useful entomological outcome, and is one that is being considered alongside epidemiological outcomes for assessing an intervention's impact by the WHO's vector control guideline committee. Statistical considerations 11.7 Analysis  • Primary Analysis and Missing Data: "Although we expect the randomisation to produce balanced covariate structures, we will consider methods of adjustment to balance baseline covariates should there be differences between the arms." – It might be helpful to pre-specify the covariates that you would expect to be balanced and consider adjustment for if found to be unbalanced (eg. ITN use could be an important covariate. I note this is mentioned on the definitions page but I don't think it is stated during the main paper that a measure of ITN use will be collected.) It is good to see both treatment and vector control interventions being considered alongside each other. Presumptive treatment can rapidly reduce the prevalence but it will return to the previous levels if the vectorial capacity is not also reduced. It will be intriguing to see over the course of the study which of the interventions have the more immediate and long term effects. This protocol was an interesting read and I am glad to have been involved in the process.
--	---

REVIEWER	Jaline Gerardin Institute for Disease Modeling, USA
REVIEW RETURNED	14-Oct-2017

GENERAL COMMENTS	This is a very exciting study and I'm really looking forward to hearing about what happens. I just have a few questions / comments:  1. How are households to be included in the response (within 500m of index case) identified? By best guess of operations team, through mapping/GPS, or another method? 2. Intervention coverage is tough to measure. It would be really nice to assess what fraction of people are typically successfully reached during the reactive component of the RACD/rfMDA: how often are people never at home when the operations team visits their home? I realize this may be too challenging to measure. However, perhaps it would be possible to record how often each exclusion criterium is met (recent treatment, previous spraying, pregnancy, etc). It would be good to know if the same households keep popping up as targets for reactive activities but get excluded due to too-recent intervention --- this would have implications for designing reactive strategies with a more intensive component. 3. At the current transmission levels in the study area, from modeling work I'm a bit concerned that the difference in effect size across RACD and rfMDA will be small to negligible. But any lessons learned will be valuable across the whole E8 region.
---

VERSION 1 – AUTHOR RESPONSE

Reviewer 1:

Background:

1.5 Drug Administration

- My understanding is that single dose PQ is provided to all Pf index cases in both arms. It is also provided to RDT-positive household members and neighbours in the RACD arm, but not presumptively to household members and neighbours in the TPE arm. It seems there is a possibility for a confounding effect of PQ (if it is hypothesised that PQ has potential to block transmission) – skewing the results in favour of RACD. I assume there is a safety-related problem with providing PQ in the TPE arm to healthy people? Perhaps this could be explained. The issue should also be acknowledged/adjusted for if possible.

The withholding of single low dose primaquine (SLD PQ) in subjects with undocumented malaria infection in the TPE (or rfMDA, reactive focal MDA) arm is due to concerns about safety [for a discussion please see Lubell et al., newly cited as Reference 40]. This approach is in accordance with Namibian MoHSS national policy, whereby SLD PQ is reserved for those individuals whose Pf infection is test-confirmed (primarily by RDT positivity), provided they do not meet PQ exclusion criteria (age < 1 year, pregnancy, breastfeeding for <1 year, or allergy).

We appreciate the reviewer's suggestion to explore the effect that SLD PQ may have on our primary outcome measure, specifically the possibility that its transmission-blocking effect may reduce incidence relatively more in RACD arms than in TPE arms due to differential RDT testing in those arms.

We have added language to the submitted manuscript ("Procedures" -> "Reactive focal mass drug administration") stating "Participants receiving rfMDA will not be given primaquine, as effects on safety remain an area of ongoing research.[40]"

We have also modified the language in the manuscript ("Statistical analysis") to read: "Additional analyses will consider adjustments for variability in intervention timing and coverage, differential administration of single low dose primaquine in RACD versus rfMDA arms, proximity to the nearest household receiving an alternate intervention, and variation in EA characteristics that differed by intervention arm, including receipt of RAVC and/or rfMDA during the pilot phase of the study."

- On that note is there an issue with PQ at this dose causing haemolytic anaemia in people with G6PD deficiency? Is this a risk that needs to be considered in the RACD arm? Perhaps for inclusion under the safety and risks section.

Please see our reply above. For individuals with documented infection (in the RACD arm) the potential benefit outweighs the safety risk, and this is reflected in WHO policy [Updated WHO policy recommendation: single dose primaquine as a gametocytocide in Plasmodium falciparum malaria, 2012] and in national policy in Namibia.

- "Anyone who is excluded from TPE receives an RDT instead". It would be useful to state how these people will be analysed, presumably they are still considered under TPE on an intention to treat basis.

We agree that this should be clarified and have updated our manuscript (“Statistical analysis”) to state “Primary analyses will follow an intention-to-treat model. Individuals will be analysed according to their randomisation arm regardless of received treatment.”

Study Site:

- It states in the background that previous trials of TPE were challenged by their high transmission settings, but it appears here the Zambezi region has been selected for being the most malarious region in the country. I thought the purpose of the study was to trial TPE in a low transmission setting so the reasons behind the choice may need slightly further clarification.

It is true that prior successful MDA programs tended to be in low transmission settings [please see Newby et al., cited as Reference 15 in manuscript]. Namibia remains such a low transmission country, albeit with in-country regional variation in malaria incidence due to factors described in our manuscript (“Study setting and trial preparations”). During the study planning stages, our investigator team worked with Namibia’s MoHSS to identify the region that, by hosting our study, would best serve the dual purposes of aiding the country’s progress toward elimination and having a suitable epidemiology to provide sufficient power to answer the study questions. Although the selected Zambezi Region does demonstrate higher incidence than other regions within Namibia, it remains a low transmission region overall when compared with other malarious settings in sub-Saharan Africa.

We have added text to the manuscript (“Study setting and trial preparations”) stating: “The Zambezi region was selected as a study site due to having a sufficiently high incidence to provide power to answer the study questions, while still representing a very low transmission epidemiology, defined by WHO as an area with an annual parasite incidence of fewer than 100 cases per 1000 population.[33]”

Study design:

5.1 Overview

- Table 2. Reactive Vector Control exclusion criteria states households sprayed by MoHSS in past 24 hours will be excluded. Does this mean houses that have been sprayed only 2 days previously will be sprayed again? It also states later in the paper that households that have received RAVC in that season will not be sprayed again; this could be added to this table.

As RAVC with Actellic®300CS and conventional IRS in Namibia (using DDT or deltamethrin) exert their insecticidal effects by distinct biochemical mechanisms, there is no absolute contraindication to a household receiving these sequentially. The rationale behind avoidance of a given household’s receipt of both conventional IRS and RAVC on the same day, is primarily to allow sufficient time for the IRS agent to dry fully; an ancillary reason is to avoid the intrusiveness toward families of applying insecticide to their household twice in the same day. As a result however, our protocol does allow, for example, a household to receive conventional IRS followed >24 hours later by RAVC.

In Table 2, we have added the exclusion criteria for households sprayed by MOHSS in the past 24 hours.

Study procedures:

6.1 Randomisation

- It states that after removing the Enumeration Areas that do not meet the inclusion criteria, there was a total of 54 EAs. I am unclear how this list increased to 56 so that the randomisation could produce equal numbers in each arm. Where did the extra two EAs come from?

We thank the reviewer for pointing this out. In our submitted manuscript (“Randomisation”) as well as Section 5.1 of the accompanying protocol (“Overview”), we refer to 56 EAs (or 14 in each arm). Section 6.1 of the protocol document (“Randomisation”) states “The study area includes 102 Enumeration Areas (EAs). Study EAs were eligible to be included in the study if they had enough geographic data to assign cases to EAs and had at least one incident case of malaria in the previous three seasons. These criteria excluded 29 areas with missing information and 19 with no incident cases, leaving 54 EAs. To have equal numbers of EAs in each study arm, the list included 56 EAs.”

This represents an error in the Protocol. While 29 EAs were in fact excluded due to missing or incomplete incidence data, only 17 EAs had consistently had zero cases in the prior three seasons and were therefore excluded as well. We will address this with the IRBs with a minor modification request.

- I assume the allocation is randomly selected using a computer randomised algorithm – it is useful to state this as describing the randomisation process is an important part of reviewing the risk of bias in a systematic review.

We appreciate this suggestion and have modified the language in the submitted manuscript (“Randomisation”) to read “...These clusters were randomly allocated by computer-generated algorithm to one of four arms using restricted randomisation to balance the distribution of key characteristics across study arms... A set of 100,000 reassignments meeting the restriction criteria was generated. A set of ten allocations meeting the criteria was randomly selected by computer algorithm. The final allocation was randomly selected by local MoHSS staff.”

- I am not sure if the interventions were applied during year 1 of the study. If so, there is a risk of contamination from the study villages that received RAVC as pirimiphos is active for up to 12 months. EAs in the non-RAVC arm may therefore be benefitting from IRS from the previous season. It may be possible to consider a measure of time since last IRS treatment as a covariate during the analysis, or discuss the limitations of the study for reviewing the impact of RAVC vs a control group that is potentially contaminated by IRS (both as a standard MoHSS procedure and in those that received RAVC in the first year of the study)

EAs were in receipt of RAVC during the pilot phase of the study in 2016, and we are unable to exclude the possibility of confounding from those interventions. We have modified the submitted manuscript (“Statistical analysis”) to read: “Additional analyses will consider adjustments for variability in intervention timing and coverage, differential administration of single low dose primaquine in RACD versus rfMDA arms, proximity to the nearest household receiving an alternate intervention, and variation in EA characteristics that differed by intervention arm, including receipt of RAVC and/or rfMDA during the pilot phase of the study.”

We feel that inter-season contamination from receipt of conventional IRS is less likely to confound the primary outcome in a systematic way, as all households in the Study Area were equally targeted by MoHSS in its yearly IRS campaign.

Entomological management:

9.2 Mosquito identification

- I note that the study does not intend to report entomological outcomes, but if mosquitoes are being identified, it would be interesting to know how these mosquitoes will be collected. If this is conducted in a standardised way eg. with CDC light traps, it would be possible to report the number of vectors per trap per night (or time period). This measure of vector density is a useful entomological outcome, and is one that is being considered alongside epidemiological outcomes for assessing an intervention’s impact by the WHO’s vector control guideline committee.

Mosquito collection is included in the protocol, yet has not been implemented in a way that is balanced across arms. As such we have not provided details in the manuscript.

Statistical considerations

11.7 Analysis

- Primary Analysis and Missing Data: “Although we expect the randomisation to produce balanced covariate structures, we will consider methods of adjustment to balance baseline covariates should there be differences between the arms.” – It might be helpful to pre-specify the covariates that you would expect to be balanced and consider adjustment for if found to be unbalanced (eg. ITN use could be an important covariate. I note this is mentioned on the definitions page but I don’t think it is stated during the main paper that a measure of ITN use will be collected.)

Thank you for this suggestion. We have modified the text (“Statistical analysis”) of the manuscript that pertains to potential further adjustments for individual level factors: “Although we expect randomisation to produce balanced covariate structures, we will consider methods of adjustment to balance baseline covariates (e.g., LLITN use, pre-transmission season IRS coverage, travel history, housing, occupation, ecological factors) should there be detected differences between the arms.”

Reviewer 2:

1. How are households to be included in the response (within 500m of index case) identified? By best guess of operations team, through mapping/GPS, or another method?

Identification of households in the Target Area is facilitated by the Spatial Decision Support System (SDSS) established prior to trial initiation. Linked to the Zambezi Region rapid reporting system, the SDSS allows study team members to identify households within any specified radius around the household of a newly diagnosed index case

We have updated the language in the submitted manuscript to be clearer on study operations’ facilitation by the SDSS. Having introduced the SDSS earlier in the manuscript (“Study setting and trial preparations”), we now specify in a subsequent section (“Enrollment of participants for study intervention”) that: “Using output from the SDSS to plan the intervention, a study team, consisting of a field investigator, nurse, and driver/data collector, will be dispatched to the home of the index case.”

2. Intervention coverage is tough to measure. It would be really nice to assess what fraction of people are typically successfully reached during the reactive component of the RACD/rfMDA: how often are people never at home when the operations team visits their home? I realize this may be too challenging to measure. However, perhaps it would be possible to record how often each exclusion criterium is met (recent treatment, previous spraying, pregnancy, etc). It would be good to know if the same households keep popping up as targets for reactive activities but get excluded due to too-recent intervention --- this would have implications for designing reactive strategies with a more intensive component.

The study teams are recording absenteeism for the study interventions, as well as excluding conditions for rfMDA. The frequency of exclusion due to too-recent intervention is also being recorded, and all these enrollment indicators will be reported to inform the design of reactive strategies.

3. At the current transmission levels in the study area, from modeling work I'm a bit concerned that the difference in effect size across RACD and rfMDA will be small to negligible. But any lessons learned will be valuable across the whole E8 region.

We agree with the reviewer that, if national and regional incidences continue their overall pattern of decline over time, the effect sizes across rfMDA and RACD, or RAVC and no RAVC, will become smaller. The investigator team factored an anticipated incremental decline in Zambezi regional incidence into the original study design, yet adjustments were then necessarily made in response to the 2016 outbreak. Despite the component of unpredictability in regional incidence and ensuing power calculations, we hope that this trial's results will be valuable for Namibia and the region, not only with regards to the effectiveness of reactive focal interventions, but also with regards to the feasibility issues (cost, acceptability, adherence, safety, coverage).

VERSION 2 – REVIEW

REVIEWER	Jaline Gerardin Institute for Disease Modeling, USA
REVIEW RETURNED	04-Dec-2017

GENERAL COMMENTS	No additional comments
------------------------

REVIEWER	Leslie Choi Liverpool School of Tropical Medicine UK
REVIEW RETURNED	04-Dec-2017

GENERAL COMMENTS	All of the referee comments have been sufficiently addressed and I have none outstanding comments.
--